# SERS Determination of Trace Phosphate in Aquaculture Water Based on a Rhodamine 6G Molecular Probe Association Reaction

**DOI:** 10.3390/bios12050319

**Published:** 2022-05-10

**Authors:** Ye Jiang, Xiaochan Wang, Guo Zhao, Yinyan Shi, Nguyen Thi Dieu Thuy, Haolin Yang

**Affiliations:** College of Engineering, Nanjing Agricultural University, Nanjing 210031, China; 2018212006@njau.edu.cn (Y.J.); zhaoguo@njau.edu.cn (G.Z.); shiyinyan@njau.edu.cn (Y.S.); nguyendieuthuy@163.com (N.T.D.T.); 2018212005@njau.edu.cn (H.Y.)

**Keywords:** surface-enhanced Raman spectroscopy, association reaction, aquaculture water, phosphate, trace detection

## Abstract

Although phosphate (Pi) is a necessary nutrient for the growth of aquatic organisms, the presence of excess Pi leads to water eutrophication; thus, it is necessary to accurately determine the content of Pi in water. A method for the determination of trace Pi in aquaculture water was developed based on surface-enhanced Raman spectroscopy (SERS) combined with rhodamine 6G (R6G)-modified silver nanoparticles (AgNPs) as the active substrate. The adsorption of R6G on the AgNP surfaces led to a strong SERS signal. However, in the presence of Pi and ammonium molybdate, phosphomolybdic acid formed, which further associated with R6G to form a stable R6G-PMo_12_O_40_^3−^ association complex, thereby hindering the adsorption of R6G on the AgNPs, and reducing the SERS intensity; this sequence formed the basis of Pi detection. The decrease in the SERS intensity was linear with respect to the Pi concentration (0.2–20 μM), and the limit of detection was 29.3 nM. Upon the application of this method to the determination of Pi in aquaculture water, a recovery of 94.4–107.2% was obtained (RSD 1.77–6.18%). This study provides an accurate, rapid, and sensitive method for the trace determination of Pi in aquaculture water, which is suitable for on-site detection.

## 1. Introduction

Phosphate (Pi), which is one of the necessary nutrients for the growth of aquatic organisms [1], is widely present in natural water systems, and is particularly abundant in aquaculture water [2]. In recent years, large amounts of Pi have been released into the aqueous ecosystem from the manufacturing and agricultural industries [3,4], and from urban domestic sewage [5], resulting in the presence of excess Pi in such water systems [6]. This can be particularly damaging to the aquatic environment due to the fact that the excess Pi can stimulate the proliferation of algae and bacteria, resulting in hypoxia or oxygen depletion, changes in the species present, increased fish mortality, reduced species diversity, and reduced levels of harvestable fish; this is known as water eutrophication [7]. Indeed, water eutrophication has become one of the major threats to the sustainable development of the aquaculture industry. Therefore, the rapid and accurate determination of the Pi content in aquaculture water and the prevention of water eutrophication are necessary to permit the development of aquaculture production.

Currently, a variety of efficient and stable detection methods have been developed for the detection and determination of Pi in different water environments. These techniques include spectrophotometry [8,9], fluorimetry [10,11], phosphorescence spectroscopy [12], electrochemical sensing [13,14], colorimetry [15,16], ion chromatography [17,18], and bio-sensing [19,20]. For example, the use of ammonium molybdate spectrophotometry [9] for the determination of Pi contents is a well-developed technique that involves the reduction of phosphomolybdic acid to molybdenum blue. However, the detection limit of this method tends to be >0.01 mg/L, and complex preprocessing procedures are required. In addition, ion chromatography and fluorescence spectroscopy are rapid and accurate techniques, but they require the use of high-precision equipment for auxiliary detection, and so are not suitable for on-site or on-line detection [21]. Furthermore, although electrochemical sensors are small, highly accurate, and exhibit excellent ion selectivities, they require complex calibration prior to use and tend to have short service lives [22]. Due to the various issues associated with these methods, the development of low cost, fast, accurate, and simple detection techniques is necessary for the determination of Pi in aquaculture water.

In this context, surface-enhanced Raman spectroscopy (SERS) involves the use of noble metal nanostructures (e.g., gold, silver, or copper) with rough surfaces, wherein the Raman spectral intensity of the detected molecules adsorbed on the surfaces are significantly enhanced, and the enhancement range can reach 4–14 orders of magnitude [23]. This technique is highly sensitive and can be used for trace and even single-molecule detection [24]. Usually, only the Raman spectral intensity of the substance to be measured is selectively enhanced. As such, when water is used as a solvent, its Raman signals are not enhanced, and so the background signal is almost negligible. These factors render the Raman analysis of substances in the aquatic environment more convenient, and on-site analysis becomes possible. Due to these advantages, SERS technology has been widely employed for the trace detection of nutrients (e.g., nitrogen [25], phosphorus [26], and calcium [27]), heavy metals (e.g., copper [28], mercury [29], and cadmium [30]), micro-plastics (e.g., polyethylene, polypropylene [31], and polystyrene [32]), and antibiotics (e.g., levofloxacin [33], enrofloxacin [34], and chloramphenicol [35]) in water systems. As a further example, Luo et al. used silver nanoparticles (AgNPs) modified with cysteine and rhodamine 6G (R6G) as a SERS-active substrate and employed the competitive reaction between cysteine and pyrophosphate on Cu^2+^ to weaken the Raman signal of R6G and indirectly determine the pyrophosphate contents in serum and urine samples [36]. In addition, Li et al. reported a system based on azo dyes formed by the reaction of 4-aminothiophenol, nitrite, and *N*-(1-naphthyl)ethylenediamine hydrochloride under acidic conditions, wherein the azo dye exhibited a high SERS signal when adsorbed on AgNPs. This system allowed the concentration of nitrite in aquaculture water to be determined by measurement of the SERS signal with a low limit of detection (LOD, i.e., 9 nM) [37]. Furthermore, Jia et al. developed a graphene nanomaterial modified with hexathiol and AgNPs as a SERS substrate for the quantitative detection of polycyclic aromatic hydrocarbons (PAHs) in seawater over a concentration range of 0.1 nM to 0.5 mM [38]. However, due to many interference factors in aquaculture water (i.e., interfering ions, temperature, dissolved oxygen, and pH), in addition to the mutual transformation between phosphorus-containing substances, the weak Raman signal of Pi in water, and the difficulty of achieving Pi adsorption on the enhanced substrate surface, there is still a lack of research into the determination of Pi in aqueous systems using SERS technology.

Thus, we herein report the use of R6G-modified AgNPs (AgNPs-R6G) as a SERS-active substrate, wherein the phosphomolybdic acid formed by the reaction between Pi and ammonium molybdate will be expected to associate with R6G to weaken the SERS signal intensity. Ultimately, we aim to develop a simple, ultrasensitive, and highly selective method for the detection and determination of trace amounts of Pi in aquaculture water, and to provide a theoretical basis and technical support for the on-site and on-line determination of Pi in this environment. Subsequently, we apply the developed assay to actual samples and compare it with the traditional Pi assay, providing comparative results with a view to promoting advances in the technologies available for water safety testing.

## 2. Materials and Methods

### 2.1. Instruments

The UV–Vis absorption spectra of the NPs and the reaction products were determined using a UVmini-1280 spectrophotometer (SHIMADZU, Kyoto, Japan). Transmission electron microscopy (TEM) images of the AgNPs were obtained using a HT7700 TEM (HITACHI, Tokyo, Japan). The Raman spectra were obtained using a portable Raman spectrometer (BWS465-785S, B&W TEK, Newark, DE, USA) equipped with a 785 nm laser. The laser power was 150 mW, and the exposure time for data collection was 5 s.

### 2.2. Reagents

A silver nitrate standard solution (AgNO_3_, 0.1 M) was purchased from Anpel Laboratory Technologies Co., Ltd. (Shanghai, China). Sodium citrate dihydrate (C_6_H_5_Na_3_O_7_·2H_2_O, 99.0%), sodium borohydride (NaBH_4_, 98%), R6G (C_28_H_31_N_2_O_3_Cl, 98%), ammonium molybdate tetrahydrate (H_24_Mo_7_N_6_O_24_·4H_2_O, 99.9%), sodium phosphate dodecahydrate (Pi: Na_3_PO_4_·12H_2_O, 99.9%), and sodium chloride (NaCl, 99.9%) were purchased from Shanghai Aladdin Biochemical Technology Co., Ltd. (Shanghai, China). Sulfuric acid (H_2_SO_4_, 98%) was obtained from Shanghai Chemical Reagent Co., Ltd. (Shanghai, China). All reagents were of analytical grade and were used without further purification. All aqueous solutions were prepared using deionized water with a specific resistance of 18.2 MΩ·cm, as obtained from a Milli-Q Advantage system (Millipore, Billerica, MA, USA).

### 2.3. Synthesis of the AgNPs

The AgNPs were prepared according to the sodium citrate reduction method reported in the literature [28,36,39] with some modifications. More specifically, the 0.1 M silver nitrate solution (118 μL) and sodium citrate (88 mg) were added to deionized water (120 mL). Subsequently, an aliquot (100 μL) of a freshly prepared 0.01 M sodium borohydride solution in water was added dropwise under magnetic stirring at room temperature (25 °C), and after further stirring for 5 min, the mixture was heated to 80 °C in a water bath, and stirred for an additional 8 h. Finally, the resulting colloidal AgNP solution was cooled to room temperature (25 °C) and stored in a refrigerator at 5 °C for later use.

### 2.4. Testing Procedure

A mixture of R6G (1 × 10^−4^ M, 10 μL), ammonium molybdate (2 × 10^−2^ M, 20 μL), H_2_SO_4_ (6 M, 20 μL), and the desired quantity of Pi were added to a 5 mL marked centrifuge tube, diluted to 0.3 mL using deionized water, and mixed well. After allowing to stand for 10 min at room temperature (25 °C), a solution of the AgNPs (1.5 mL) and a 1 M solution of NaCl (50 μL) were added, and the mixture was diluted to 2.0 mL with deionized water and shaken. Finally, the mixture was allowed to react for 10 min and then transferred to a quartz cell with a 1 cm pathlength for SERS detection. Each sample was tested 5 times, and the SERS peak intensity (I_Pi_) at 1508 cm^−1^ was measured. The peak intensity of the blank (I_0_, i.e., without Pi) was also measured, and the variation in the SERS peak intensity was calculated by the formula ∆I = I_0_ − I_Pi_ to allow the quantitative calculation of Pi.

### 2.5. Sample Preparation

Four different water samples (i.e., pond aquaculture water, Yangtze River water, aquaponics water, and tap water) were employed to validate the detection accuracy of the described method. The pond aquaculture water was obtained from an aquaculture base of Nanjing Agricultural University, the Yangtze River water was taken from the vicinity of the Pukou Wharf in Nanjing City, the aquaponics water sample was obtained from the aquaponics system installed in a greenhouse on the roof of Boyuan Building, Pukou Campus, Nanjing Agriculture University, and the tap water originated from the Nanjing city water supply. The water samples were pretreated by filtration through a 0.45 μm microporous membrane, and a known amount of Pi was added to the water samples to obtain the recovery, which was used to evaluate the feasibility of the detection method.

## 3. Results and Discussion

### 3.1. Detection Principle

The AgNPs prepared using the sodium citrate reduction method were able to be stably dispersed as a colloid in solution. In terms of the detection principle, following the addition of NaCl and R6G, the AgNPs aggregated with Cl^−^ [40], which excited the surface plasmon resonance between AgNPs and triggered the enhancement of the local electromagnetic field. Subsequently, this led to the formation of multiple Raman hot spots between adjacent AgNPs particles [41]. A large number of R6G molecules are then adsorbed on the surfaces of AgNPs by electrostatic interactions, and these molecules enter the Raman hot spot positions following AgNP aggregation to produce strong SERS signals [42]. In an acidic medium, Pi and ammonium molybdate react to form phospho-molybdic acid [43], which subsequently associate with R6G to form the R6G-PMo_12_O_40_^3−^ association complex. This complex is not electrically active in solution and does not easily adsorb onto the AgNP surfaces, thereby resulting in a poor SERS activity and a decrease in the SERS signal intensity attributed to R6G. When the concentration of Pi increases, the number of R6G probe molecules in the hot spot positions decreases, and the intensity of the SERS peak decreases linearly; overall, this constitutes a new quantitative SERS method for the detection of trace Pi, as outlined in Figure 1.

### 3.2. SERS Detection of Pi

To verify the feasibility of the Pi detection method, the various combinations of the system components were analyzed separately by Raman spectroscopy (Figure 2). As shown, no obvious Raman peaks were observed for R6G (Figure 2a), the AgNPs (Figure 2b), or the AgNPs + R6G combination (Figure 2c). However, following the addition of NaCl, several Raman peaks appeared (AgNPs + R6G + NaCl, Figure 2e), indicating that the addition of NaCl could increase the number of Raman active sites on the surfaces of the AgNPs to produce numerous Raman hot spots. Among these peaks, those located at 1085, 1125, and 1182 cm^−1^ originated from the in-plane bending of the C–H bond [44], while that at 1312 cm^−1^ was attributed to the stretching vibration of the C–O–C bond [45], and the peaks at 1363, 1508, and 1650 cm^−1^ originated from the stretching vibration of the C=C bond in the benzene ring [46]. Upon the separate addition of Pi (AgNPs + R6G + NaCl + Pi, Figure 2f) and ammonium molybdate (AgNPs + R6G + NaCl + MoO_4_^2−^, Figure 2g), no significant changes in the SERS peak intensities were observed, while the simultaneous addition of Pi and ammonium molybdate significantly decreased the SERS peak intensities throughout the spectrum (AgNPs + R6G + NaCl + Pi + MoO_4_^2−^, Figure 2d). This observation indicated that the phosphomolybdic acid formed from the reaction between Pi and ammonium molybdate can associate with R6G to hinder the adsorption of R6G onto the AgNP surfaces, thereby confirming that the hypothetical working principle of this detector is feasible.

### 3.3. Characterization of the AgNPs-R6G System

The AgNPs synthesized using the sodium citrate reduction method were bright yellow in appearance and exhibited a narrow and strong absorption peak at 395 nm (Figure 3A(a)), which indicated that the AgNPs were relatively uniform and well dispersed [47,48]. As shown in Figure 3A(b), the addition of R6G did not lead to AgNP aggregation or a change in the colloidal color, although the surface plasmon band in its absorption spectrum was redshifted from 395 to 397 nm due to the decrease in the plasma oscillation frequency around the AgNPs upon the formation of R6G-modified AgNPs (AgNPs-R6G). Subsequently, the surface morphology and particle size of the AgNPs-R6G system were characterized by TEM, and it was found that the AgNPs-R6G species was well dispersed with a particle size of ~20 nm (Figure 3B). The addition of NaCl to this system resulted in an obvious color change, from bright yellow to grey (Figure 3A(c)). Furthermore, the intensity of the absorption peak at 397 nm decreased, and a new absorption peak appeared close to 725 nm. Further TEM investigations showed that in the presence of NaCl, aggregation of the AgNPs-R6G took place to form irregular large clusters (Figure 3C).

As outlined in Figure 4A, in a solution containing 60 mM H_2_SO_4_ and 25 mM NaCl, R6G adsorbed onto the AgNP surfaces through electrostatic interactions and entered the Raman hot spot positions upon aggregation of the AgNPs, thereby forming several clear SERS peaks (Figure 4). Long-term SERS tests were then carried out for the AgNPs-R6G system (Figure 4A), and the variation in the SERS intensities of the three strongest Raman peaks (i.e., 1312, 1363, and 1508 cm^−1^) were recorded over 16 days; all peaks were normalized relative to the intensity of the Raman peak at 1508 cm^−1^ (Figure 4B). As indicated, this system exhibited a good stability, with relative standard deviations (RSDs) of RSD_1312_ = 5.74%, RSD_1363_ = 5.76%, and RSD_1508_ =5.66%, for the three peaks. In addition, 20 different locations of the AgNPs-R6G solution were randomly selected for SERS evaluation (Figure 5), and it was found that the sample was uniform in nature. Again, the peaks were normalized relative to the Raman peak intensity at 1508 cm^−1^, and the peak RSDs were calculated as follows: RSD_1312_ = 2.69%, RSD_1363_ = 2.67%, and RSD_1508_ = 2.63%. Combination of the results presented in Figure 4 and Figure 5 reveal that the AgNPs-R6G system exhibited a good stability, repeatability, and uniformity. Furthermore, due to the fact that the Raman peak at 1508 cm^−1^ was the most intense and exhibited the smallest RSD, all subsequent experiments employed this peak to analyze the detection results.

### 3.4. UV-Vis Characterization of R6G-PMo_12_O_40_^3−^

In an acidic medium (pH ~1.18), R6G exhibits a strong absorption peak at 535 nm, as can be seen in Figure 6. Following the addition of a fixed concentration of ammonium molybdate and different concentrations of Pi to R6G, the intensity of the absorption peak at 535 nm gradually decreased with an increasing Pi concentration, and a new absorption peak appeared at 562 nm. This indicates that the electrons in the generated phosphomolybdic acid conjugated with the electrons in R6G to form a stable R6G-PMo_12_O_40_^3−^ association complex. The binding force that drives this association mainly originates from the positive charge of R6G and the negative charge of the phosphomolybdic acid counter-anion, in addition to the hydrophobic forces between the R6G molecules [49].

### 3.5. Method Optimization

#### 3.5.1. Effect of the R6G Concentration

In general, the accuracy of the developed method depends on whether the R6G molecules involved in the association reaction can be reflected by variation in the intensity of the SERS peak. It must therefore be ensured that all R6G molecules are adsorbed onto the AgNP surfaces and that they enter the Raman hot-spot positions to improve the sensitivity of the detection system. Thus, the results of SERS detection on the AgNPs substrates in the presence of different concentrations of R6G (0.1–1 μM) are shown in Figure 7A. As indicated, the intensity of the SERS peak at 1508 cm^−1^ (Figure 7B) approached its maximum value at an R6G concentration of 0.5 μM, beyond which point, only a very gradual increase in intensity was observed upon increasing the R6G concentration. This result indicates that all Raman hot spots present on the AgNP surfaces were occupied by R6G molecules at an R6G concentration of 0.5 μM.

#### 3.5.2. Effect of the NaCl Concentration

The aggregation of AgNPs is known to affect the coupling of electromagnetic fields, which are essential for enhancing the Raman signal. Since the concentration of NaCl directly affects the aggregation degree of the AgNPs [41], which in turn affects the number of Raman hot spots generated by the substrate, the influence of the NaCl concentration between 5 and 50 mM was investigated to optimize the number of Raman hot spots and improve the detection sensitivity (Figure 8A). As indicated, upon increasing the NaCl concentration, the SERS peak intensity at 1508 cm^−1^ initially increased, reaching a maximum at 25 mM, and then subsequently decreased once again (Figure 8B). This trend indicated that the AgNP substrate generated the greatest number of Raman hot spots at a NaCl concentration of 25 mM.

#### 3.5.3. Effect of the Ammonium Molybdate Concentration

As described above, in an acidic medium, ammonium molybdate and Pi react to form phosphomolybdic acid prior to undergoing an association reaction with R6G [43]. As such, the concentration of ammonium molybdate will have a significant effect on this reaction and on the subsequent association process. Thus, to investigate the effect of the ammonium molybdate concentration on the change in SERS peak intensity (ΔI) at 1508 cm^−1^ (i.e., before and after the addition of Pi as the optimization target), SERS was carried out at a range of ammonium molybdate concentrations (0.05–0.3 mM) and with a Pi concentration of 20 μM when desired (Figure 9A). As shown in Figure 9B, ∆I initially increases with an increasing ammonium molybdate concentration prior to decreasing, and a maximum was observed at 0.2 mM. Subsequent experiments were therefore carried out using an ammonium molybdate concentration of 0.2 mM.

### 3.6. Analytical Performance of the Pi Detection Method

The variation in the SERS intensity at different Pi concentrations was then tested under the optimal experimental conditions, wherein it can be seen from the results presented in Figure 10A that the SERS intensity gradually decreased upon increasing the Pi concentration. To quantify the relationship between the SERS intensity and the Pi concentration, a standard curve was established between the Pi concentration (C_Pi_) and the change in intensity (ΔI) of the characteristic peak at 1508 cm^−1^ (Figure 10B). The curve fitting results showed a good linear relationship between ΔI and the C_Pi_ over a concentration range of 0.2–20 μM, with a correlation coefficient of R^2^ = 0.9975 and a limit of detection (LOD, (3SD_I0_/slope, where SD_I0_ is the standard deviation of I_0_, and the slope is 288.28) of 29.3 nM being obtained at a signal-to-noise ratio of 3. Table 1 lists the corresponding values of some previously reported methods for the determination of Pi, and based on these results it is apparent that our SERS-based is highly sensitive over a relatively wide concentration range.

### 3.7. Selectivity of the Developed Method

Considering that this method is targeted for use in the detection and determination of Pi in aquaculture water, it was necessary to evaluate its selectivity in the presence of common anions and cations that can be found in aquaculture water. Thus, Figure 11 shows a comparison of the detection results obtained for some select coexisting substances in deionized water at a concentration of 200 μM, and for Pi at a concentration of 20 μM. It was found that our system exhibited a good sensitivity toward Pi, since the addition of various interfering ions did not cause any significant variation in ΔI, and so within a certain concentration range our results indicate that the common anions and cations present in aquaculture water should not interfere with the detection of Pi.

### 3.8. Determination of Pi in Real Samples

To verify the reliability of this method for the determination of Pi in actual aquaculture water, different concentrations of Pi were added to pretreated pond aquaculture water, Yangtze River water, aquaponics water, and tap water, and the samples were analyzed under the optimal experimental conditions. More specifically, the phosphate contents of the pond aquaculture water, Yangtze River water, and tap water samples were all detected within normal levels. However, the phosphate content of the aquaponics water sample was determined to be slightly higher than the spiked level. This was attributed to the fact that this sample was taken from the aquaponics system in the greenhouse, and that the breeding density is higher than that of the traditional breeding method. The specific test results obtained for the four water samples are presented in Table 2; the recoveries were in the range of 94.4–107.2% with RSDs of 1.77–6.18%, thereby indicating that this method can be effectively used for the detection and determination of Pi in aquaculture water. In addition, our method was also compared with the determination of Pi using the spectrophotometric approach [53], and relatively consistent results were obtained. These observations therefore indicate that our developed method has a high detection accuracy and retains its high selectivity toward Pi in the complex aquaculture water environment.

## 4. Conclusions

In this study, a low cost, ultrasensitive, and highly selective method for the determination of trace phosphate (Pi) in aquaculture water was designed using surface-enhanced Raman spectroscopy (SERS) combined with rhodamine 6G (R6G)-modified silver nanoparticles (AgNPs) as the active substrate. Our results showed that in an acidic medium, the phosphomolybdic acid formed by the reaction between Pi and ammonium molybdate associated with R6G, which inhibited the adsorption of R6G onto the surface of the AgNPs, and led to a decrease in the SERS intensity of the R6G molecules. As a result, the detection and determination of Pi was achieved by measuring this decrease in the SERS intensity. Importantly, over a Pi concentration range of 0.2–20 μM, our method exhibited a good linearity (R^2^ = 0.9975) in terms of the variation in the SERS intensity, and a low detection limit of 29.3 nM was achieved. Compared with the spectrophotometric approach, our method is more applicable to the detection and determination of Pi in low concentration and complex water environments. In addition, this method was combined with a portable Raman spectrometer to allow the rapid on-site detection of Pi in natural water samples. Furthermore, we expect that due to the ongoing progress of flow injection technology, the combination of our method with reasonable auxiliary detection equipment will render it possible to carry out the SERS-based on-line determination of Pi, and this represents one technical focus of our future research.

## Figures and Tables

**Figure 1 biosensors-12-00319-f001:**
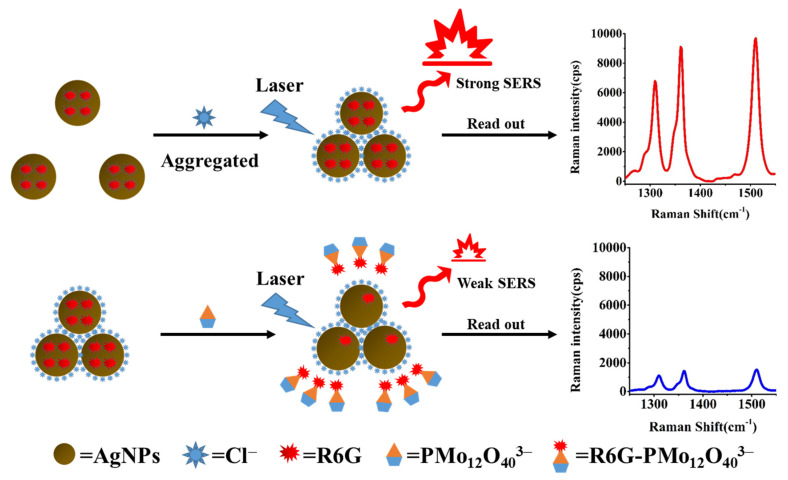
The principle of the AgNPs/Rh6G SERS nano-sensor for the detection of Pi.

**Figure 2 biosensors-12-00319-f002:**
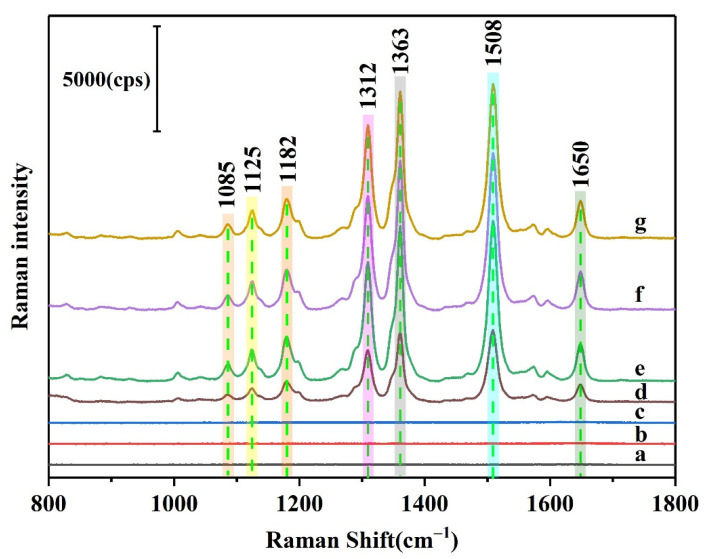
SERS spectra of combinations of the various system components: (**a**) R6G, (**b**) AgNPs, (**c**) AgNPs + R6G, (**d**) AgNPs + R6G + NaCl + Pi + MoO_4_^2−^, (**e**) AgNPs + R6G + NaCl, (**f**) AgNPs + R6G + NaCl + Pi, and (**g**) AgNPs + R6G + NaCl + MoO_4_^2−^.

**Figure 3 biosensors-12-00319-f003:**
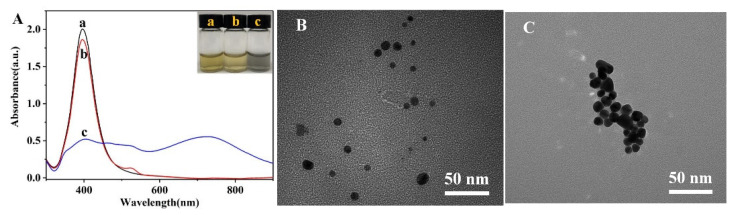
(**A**) UV-vis spectra and photographic images of (**a**) the unmodified AgNPs, (**b**) the R6G-modified AgNPs, and (**c**) the NaCl-induced aggregates of the R6G-modified AgNPs. TEM images of (**B**) the R6G-modified AgNPs and (**C**) the NaCl-induced aggregates of R6G-modified AgNPs.

**Figure 4 biosensors-12-00319-f004:**
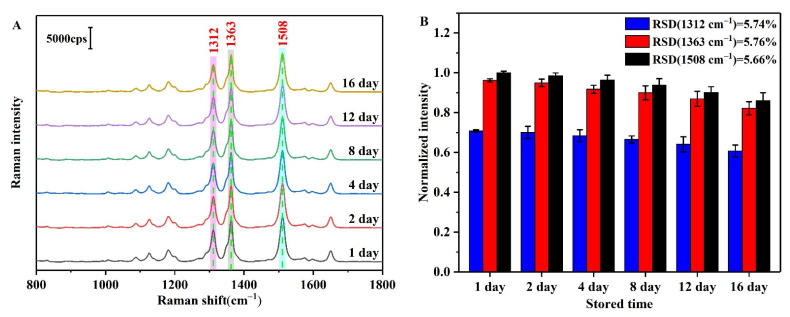
(**A**) SERS spectra of the AgNPs-R6G system stored over 1–16 days. (**B**) Normalized peak intensities at 1312, 1363, and 1508 cm^−1^.

**Figure 5 biosensors-12-00319-f005:**
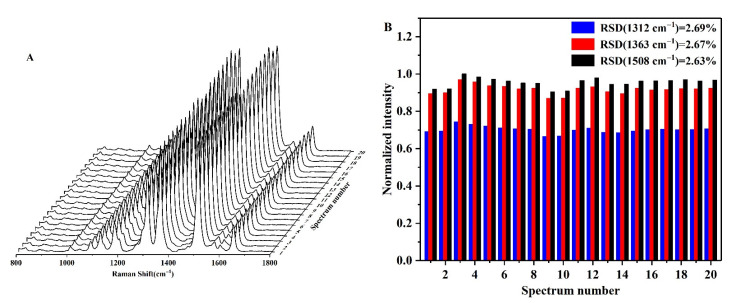
(**A**) SERS spectra of 20 locations randomly selected from the AgNPs-R6G system. (**B**) Normalized peak intensities at 1312, 1363, and 1508 cm^−1^.

**Figure 6 biosensors-12-00319-f006:**
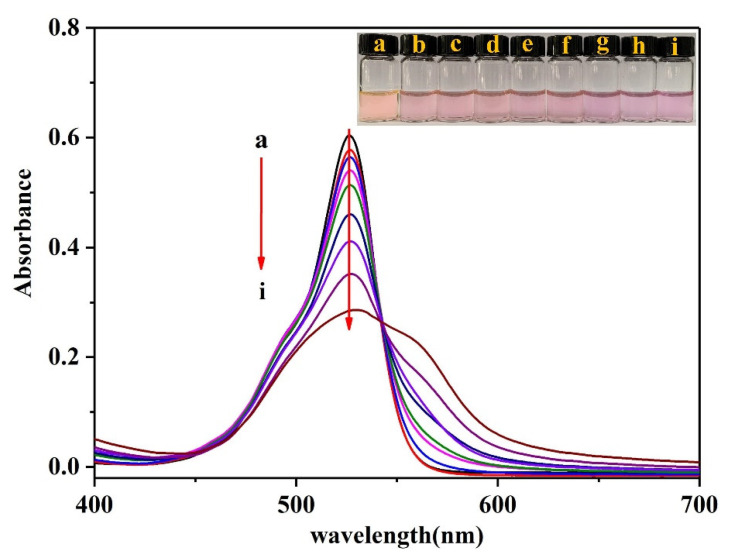
UV-Vis spectra of the Rh6G-ammonium molybdate-Pi system. (**a**) 60 mM H_2_SO_4_ + 0.5 μM R6G + 25 mM NaCl; (**b**) a + 0.2 mM ammonium molybdate; (**c**) b + 0.8 μM Pi; (**d**) b + 2 μM Pi; (**e**) b + 4 μM Pi; (**f**) b + 8 μM Pi; (**g**) b + 12 μM Pi; (**h**) b + 16 μM Pi; and (**i**) b + 20 μM Pi.

**Figure 7 biosensors-12-00319-f007:**
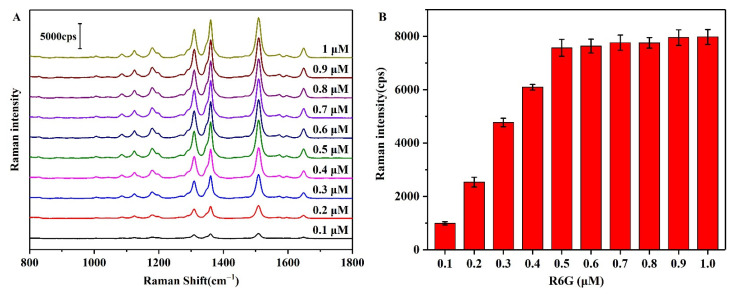
(**A**) SERS spectra of the AgNPs-R6G aggregates in the presence of different concentrations of R6G. (**B**) Intensity of the SERS peak at 1508 cm^−1^ in the presence of different concentrations of R6G.

**Figure 8 biosensors-12-00319-f008:**
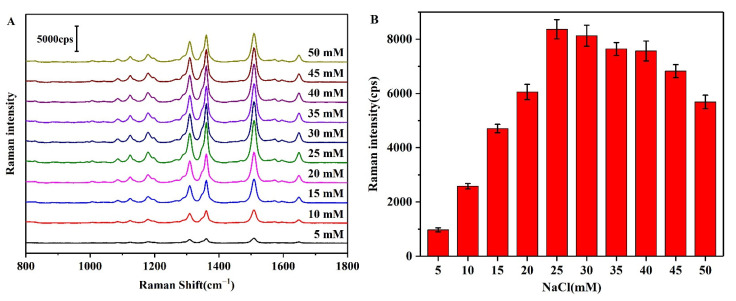
(**A**) SERS spectra of the NaCl-induced AgNPs-R6G aggregates in the presence of different concentrations of NaCl. (**B**) Intensity of the SERS peak at 1508 cm^−1^ in the presence of different concentrations of NaCl.

**Figure 9 biosensors-12-00319-f009:**
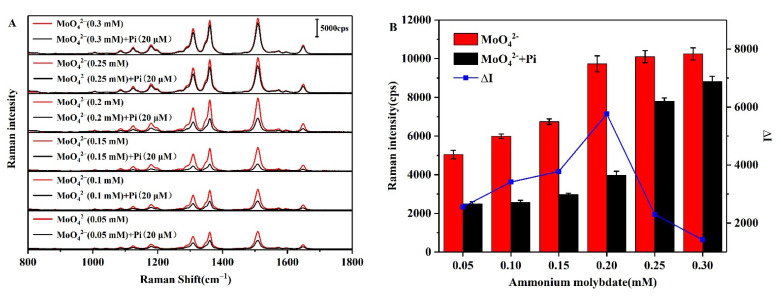
(**A**) SERS spectra and (**B**) ∆I values for the AgNPs-R6G system in the presence and absence of Pi, and upon the addition of different concentrations of ammonium molybdate.

**Figure 10 biosensors-12-00319-f010:**
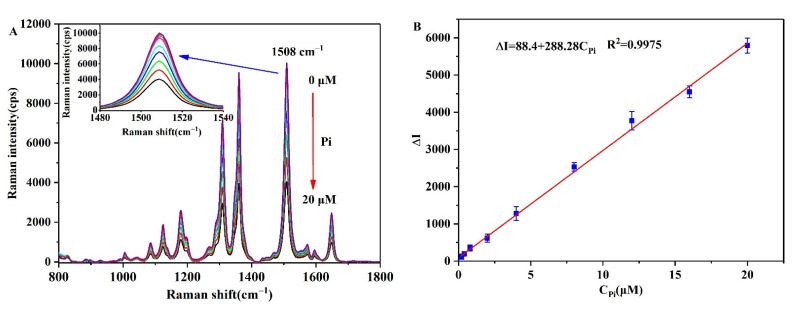
(**A**) SERS spectra of the NaCl-induced AgNPs-R6G aggregates in the presence of different concentrations of Pi. (**B**) The calibration curve of Pi based on ΔI.

**Figure 11 biosensors-12-00319-f011:**
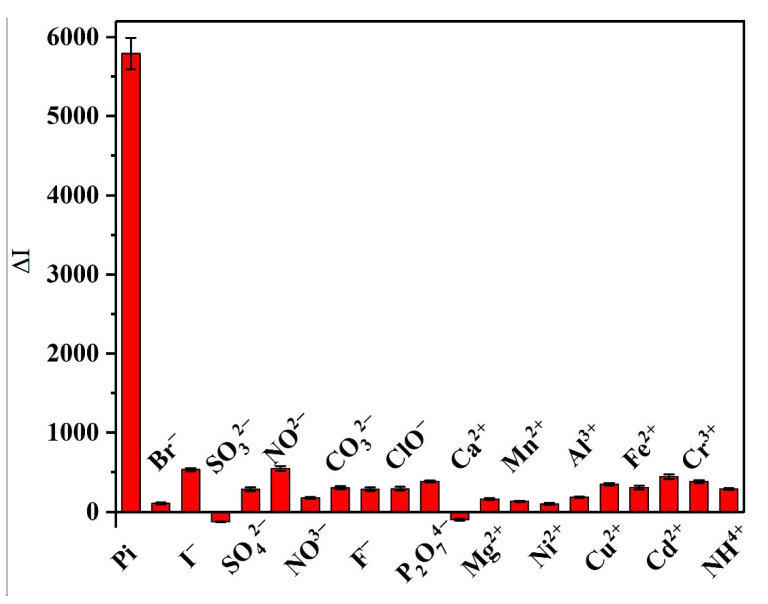
Effects of various interfering anions and cations on the SERS intensity during the detection of Pi.

**Table 1 biosensors-12-00319-t001:** Comparison of previously reported methods with our developed method for the detection and determination of Pi.

Method	Sample	Linear Range	LOD	Reference
Fluorimetry	Drinking water	0–10 μM	54 nM	[10]
Phosphorescence	Environmental water	8–320 mM	2.71 mM	[12]
Chromatography	Reservoir water	0–100 μg/L	0.7 μg/L	[18]
Biosensor	Water	248–1456 μM	45 μM	[19]
Colorimetry	River water	0.5–30 μM	76 nM	[50]
Fluorimetry	River water	7–30 μM	50 nM	[51]
Electrochemistry	Natural water	0–0.045 mg/L	3.01 μg/L	[52]
SERS	Aquaculture water	0.2–20 μM	29.3 nM	This work

**Table 2 biosensors-12-00319-t002:** Determination of Pi in aquaculture water samples by the present method and using the spectrophotometric approach.

Sample	Added Pi (μM)	This Work			Spectrophotometry (μM)
		Found (μM)	Recovery (%)	RSD (%)	
Pond aquaculture water	0	1.91		2.66	2.03
	5	6.63	94.4	5.06	7.13
	10	11.58	96.7	4.52	12.09
Yangtze River water	0	2.14		5.03	2.32
	5	7.05	98.2	3.07	7.27
	10	12.26	101.2	2.62	12.62
Aquaponics water	0	6.99		5.78	7.13
	5	12.1	102.2	1.77	12.02
	10	16.68	95.5	2.65	17.24
Tap water	0	0.79		6.18	0.71
	5	6.15	107.2	3.55	5.93
	10	10.83	100.4	2.41	10.55

## Data Availability

Not applicable.

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
