# Peer review of "SERS Determination of Trace Phosphate in Aquaculture Water Based on a Rhodamine 6G Molecular Probe Association Reaction"

_biosensors, 2022, doi:10.3390/bios12050319_

Round 1

Reviewer 1 Report

This is a routine work, but is well organized.

some comments for author to consider

a) The introduction section should be enhanced to illustrate why this study is novel and stage its significance 

b) Performance comparison must be done with other similar studies in the field. 

c) Mechanism explanation should be enhanced either with more experimental results or citing others

Author Response

Dear Editors and Reviewers,

      Thank you for your letter and for the reviewers’comments concerning ourmanuscript entitled“SERS determination of trace phosphate in aquaculture water based on a rhodamine 6G molecular probe association reaction”(ID: biosensors-1704680). Those comments are all valuableand very helpful for revising and improving our paper,as well as the important guiding significance to our researches. We have studied comments carefully and havemade correction which we hope meet with approval. I have included the responses to each comment and the revised manuscript in the attachment.

Reviewer 2 Report

The authors present a manuscript on detection of phosphate (Pi) in aquaculture water based on SERS spectroscopy by employing AgNPs conjugated with rhodamine as SERS-active substrate. The work is interesting, and the experimental approach is valid. Nevertheless, there are some questions that need to be addressed to accept the manuscript for publication.

A list of comments is reported below:

  1. line 61-62: “Since water molecules create negligible background signals due to their lack of Raman activity,…”. This statement is incorrect, water molecules do not lack of Raman activity. Raman spectrum of water has major contributes from the symmetric and asymmetric stretching of hydroxyl groups in the region 3000-3500 cm-1. Instead, in the fingerprint region (800-1800 cm-1), the Raman spectrum of water do not show intense peaks that might overlap with the spectrum of the analytes of interest. Please correct this sentence.
  2. line 216, how have the authors determined the relative standard deviations for the three Raman peaks?

  3. Figure 4B and Figure 5B report the normalized peak intensity. The authors should report with respect to which Raman peak the normalization is performed.

  4. The authors report that an acidic medium is needed for the reaction of ammonium molybdate and Pi that leads to the formation of phosphomolybdic acid. At which pH values are performed the experiments reported in the manuscript? This information should be reported.

  5. How is the LOD calculated? Please report the equation employed fro the calculation in the manuscript.

  6. How are the error bars of Figure 10B calculated?

  7. Concerning the selectivity of the method (section 3.7) it is not clear if the authors added both Pi and the ions in the same water sample, or if they added Pi and ions in different water samples.

  8. In section 3.8, what do the authors mean by “recovery”? How is this parameter calculated? How can recovery by greater than 100%?

  9. How is the RSD (%) of Table 2 calculated?

  10. How the spectrophotometry experiments to determine the concentration of Pi were performed?

  11. From the results reported in Table 2, the proposed method for the detection of Pi seems to overestimate the actual concentration of Pi in water sample. This aspect should be discussed in the manuscript.

Author Response

Dear Editors and Reviewers,

    Thank you for your letter and for the reviewers’comments concerning ourmanuscript entitled“SERS determination of trace phosphate in aquaculture water based on a rhodamine 6G molecular probe association reaction”(ID: biosensors-1704680).Those comments are all valuableand very helpful for revising and improving our paper,as well as the important guiding significance to our researches. We have studied comments carefully and havemade correction which we hope meet with approval. I have included the responses to each comment and the revised manuscript in the attachment.

Round 2

Reviewer 1 Report

The revised manuscript is ready for publication 

Author Response

Dear Reviewer

Thank you very much for your work in revising the manuscript, and on behalf of all the co-authors, I would like to express my most sincere respect to you.

Thank you and best regards.

Yours sincerely,

Ye Jiang